
# Modeling and evaluation of the susceptibility to landslide events using machine learning algorithms in the province of Chañaral, Atacama region, Chile

Francisco Parra[1], Jaime González[2], Max Chacón[1], and Mauricio Marín[1]

[1]Department of Informatics, Universidad de Santiago de Chile, Santiago, Chile
[2]Department of Geology, Universidad de Chile, Santiago, Chile

**Correspondence:** Francisco Parra (francisco.parra.o@usach.cl)

**Abstract.** Landslides represent one of the main geological hazards, especially in Chile. The main purpose of this study is to evaluate the application of machine learning algorithms (SVM, RF, XGBoost and logistic regression) and compare the results for the modeling of landslides susceptibility in the province of Chañaral, III region, Chile. A total of 86 sites are identified using various sources, plus another 86 sites as non-landslides, which are randomly divided, and then a cross-validation process is applied to calculate the accuracy of the models. After that, from 23 conditioning factors, 12 were chosen based on the information gain ratio (IGR). Subsequently, 5 factors are excluded by the correlation criterion, of which 2 that have not been used in the literature (NDGI, EVI) are used. The performance of the models is evaluated through the area under the ROC curve (AUC). To study the statistical behavior of the model, the Friedman non-parametric test is performed to compare the performance with the other algorithms and the Nemenyi test for pairwise comparison. Of the algorithms used, the RF (AUC = 0.9095) and the SVM (AUC = 0.9089) has the highest accuracy values measured in AUC compared to the other models and can be used for the same purpose in other geographic areas with similar characteristics. The findings of this investigation have the potential to assist in land use planning, landslide risk reduction, and informed decision making in the surrounding zones.

## 1 Introduction

Geological hazards constitute one of the greatest impacts that global and local economies, as well as human settlements, may face (Li et al. (2020), Wang et al. (2019)). In particular, landslides, which are defined as movements of soil, mud, debris, or rock, are the most common geological hazard in the world (Dang et al., 2020). Landslides are commonly induced by natural events such as earthquakes or heavy rainfall, which occur in specific geological, geomorphological, and hydrological environments. Nevertheless, other key factors that influence the mechanisms of landslide failure, such as in situ stresses, weathering, and heave, are also crucial. In mountainous areas, landslides can have significant effects on the topographic features, forests, soil properties (such as consistency, structure, density, and temperature), as well as on infrastructure such as roads and farming land. The extent of these effects will depend on the magnitude of the landslides (Cao et al. (2019), Saha et al. (2021)) . In the last two decades, efforts in assessing landslides have focused on studying susceptible zones, understanding the mechanisms that govern landslides (Tien Bui et al. (2017), Pourghasemi and Rahmati (2018)). This has made it possible to extract valuable knowledge


from the analysis of geomorphological, tectonic, geological, climatic, and anthropomorphic characteristics (Bui et al. (2020),
Gorsevski et al. (2006), Flentje et al. (2007), Hervás and Bobrowsky (2009)). In Chile, landslides are one of the most important
geological hazards, together with earthquakes, volcanic activity and floods. The geological, geomorphological, tectonic and
climatic conditions of the country, characterized by the presence of the Andes Mountain Range on its eastern margin and the
Coastal Mountain Range on its western margin, make it highly susceptible to the generation of mass movements, such as
landslides and rockfalls, flows and falls (Lara and Sepúlveda, 2010). Within 52 declared events in Chile, there is a total of 1010
fatal victims caused by landslides, corresponding to 882 deaths and 128 people missing between 1928 and 2017 (90 years)
(Marín et al., 2018).

The Atacama Region is characterized by a geomorphology that renders it susceptible to landslide events. According to
documented records, this region has witnessed the highest number of fatalities resulting from flow-type events in the country,
with a total of 132 people (Marín et al., 2018). In the area of study, the Salado River basin, situated in the Chañaral Province
of the region, a landslide event occurred in 1940 that resulted in the destruction of houses and interruption of roads, causing
substantial disruption to the city of Chañaral. In 1972, another landslide affected Chañaral and towns upstream, which were
flooded and 700 people were affected in Chañaral and 400 people in El Salado. Another flood in 1983 affected the city due to a
rise in the river level caused by an increase in rainfall (González, 2018). In Chañaral, there have been at least 15 major landslide
events in the last 150 years (Vargas Easton et al., 2018). With regard to the reports of deadly events and economic damage
provoked by landslides, the identification of areas prone to these types of events and the determination of their risk level are the
most critical actions in the assessment of the hazard (Abedini et al., 2019a). In the last two decades, extensive research has been
carried out on landslide methods using innovative technologies and tools to promote this field, such as in crisis management in
mountain areas or near them (Dahal et al., 2008). Therefore, in order to obtain a reliable and accurate vulnerability map, it is
necessary to test and evaluate several quantitative methods for more effective management of mountainous areas Abedini et al.
(2019a). The use of Geographical Information Systems (GIS) in the elaboration of susceptibility maps constitutes an effective
method to identify and delineate landslide-prone areas. This allows for the creation of a geospatial database of occurrences
or an exhaustive inventory. By using GIS data repositories, the geospatial attributes of landslide-prone sites that can influence
the potential stability of slopes, called landslide conditioning factors (LCFs), can be aggregated into a database. These include
factors such as slope aspect and angle, precipitation, soil types, lithologies, and others. Merghadi et al. (2020).

Several methodologies and techniques have been developed for the hazard susceptiblity cartography around the world.
Literature has classified them into (Abedini et al. (2019b), Merghadi et al. (2020)):

1. Models founded on physical conditions

2. Models founded on expert knowledge (Shirzadi et al. (2012), Zhang et al. (2016)).

3. Multivariate statistical methods. The examples are the statistical index (SI) Regmi et al. (2014), the frequency ratio (FR)
(Pham et al. (2015), Shirzadi et al. (2012)), and the logistic regression (Shirzadi et al. (2012), Tsangaratos and Ilia (2016),
Chen et al. (2019), Mousavi et al. (2011)).



4.  Machine learning models, such as decision trees (DT) (Khosravi et al., 2018), random forest (RF) (Hong et al., 2016), artificial neural networks (ANN) (Pham et al. (2018), Shirzadi et al. (2012)) and some hybrid methods which include optimization algorithms (Abedini et al. (2019a), Ahmadlou et al. (2019), Tien Bui et al. (2017), Chen and Guestrin (2016)).

.

Each of the methods described has unique strengths and limitations (Khosravi et al., 2018). Physical models, for example, require extensive field analysis and are currently considered unbeatable in terms of prediction accuracy, making them suitable for local scale mapping. In order to work effectively, these models demand a complete knowledge of the landslide systems, obtained through meticulous observation and monitoring of the surface and the subsurface; this is essential in order to issue timely warnings of further slope collapse (Piciullo et al., 2018). However, when applied on a larger scale, the need for a large amount of substantial data to obtain reliable results becomes inconvenient due to the considerable financial and computational resources required. Therefore, the use of this technique for the segmentation of larger regions is not feasible. This has led to the proliferation of statistical and knowledge-based models for more than four decades (Guzzetti et al., 2012). The knowledge-based models operate on the premise of building a framework with limited information, which is then parameterised by a system of weights assigned to factors according to expert judgement. Statistical models, on the other hand, have benefited from recent advances in GIS. This has paved the way or the beginning and the successful application of a set of tools and cuantitave methodologies for the modelling of landslides, improving in this way the understanding of the associated patterns and the causative agents (Dou et al., 2019). In the last twenty years, several susceptibility models derived from statistical approaches have found application in machine learning to produce risk zonation maps. At the moment, the thin line that separates the statistical models from machine learning is a subject of controversy (Ij (2018), Merghadi et al. (2020)). The synergy and differences between statistical methods and machine learning are not clearly explained in academic works, mainly because the approach for geoscientists is primarily to generate and refine accurate results in landslide susceptibility mapping (LSM) rather than algorithmic categorisation. In essence, machine learning is characterised by its ability to extract knowledge from data without relying on rule-based functions, whereas statistical modelling aims to establish relationships between data variables through algebraic expressions. Although the two fields were once considered mutually exclusive, they have recently converged (Merghadi et al., 2020). A notable example is the adoption of the logistic regression (LR) algorithm, originally from statistics, to solve classification problems. Now machine learning has adopted LR and has become one of the most widely used algorithms. However, machine learning is more concerned with optimisation and efficiency, in contrast to the inferential approach of statistical models.

Machine learning methods have been used in engineering and science problems for more than two decades. This is the reason why the use of these techniques in the area of geosciences and remote sensing is quite new and limited. Machine learning focuses on the automatic extraction of information from data through computational and statistical methods. The areas of applicability are very diverse, and involve different topics such as rock mass characterization, ocean products, vegetation indices, etc. At present, data analysis methods play a central role in geosciences and remote sensing. While collecting large volumes of data is essential in the field, and the analysis of this information becomes a major challenge (Lary et al., 2016).





Various machine learning techniques, including random forest (RF), support vector machine (SVM), and artificial neural network (ANN), have proven to be effective in dealing with nonlinear data across different scales in areas such as identification, prediction, mitigation, and modeling. Studies like Sekkeravani et al. (2022), Miao et al. (2018), Saha et al. (2022), Conforti et al. (2014) and others have demonstrated the success of these methods. Unlike traditional statistical models, which aim to infer relationships between variables, machine learning models autonomously identify logical criteria from input data to make highly accurate predictions (Miao et al., 2023). The main advantages of the machine learning are the following:

– **Increased accuracy**: Machine Learning models can learn complex, nonlinear patterns in the data, which can increase the accuracy of predictions. In addition, Machine Learning models can handle large amounts of data and variables, which can capture a wide variety of factors that can influence landslide susceptibility.

– **Flexibility**: Machine Learning models are highly flexible and can be adapted to different types of data and problems. This allows researchers to use different modeling techniques and explore different variables and combinations of variables to find the best solution.

– **Speed**: Machine Learning models can process large amounts of data quickly and efficiently. This can save time and resources compared to traditional methods that may require a manual and tedious process of data analysis.

– **Generalization**: Machine Learning models have the ability to generalize to new situations and data. This means that, once the model is trained, it can be applied to new data and produce accurate and reliable predictions.

However, there are several research gaps in the modeling of landslide susceptibility using machine learning algorithms. Some of them are:

– **Integration of multiple factors**: Most studies have focused on assessing a limited and repetitive set of factors. The integration of new factors needs to be explored to improve model accuracy and ensure a more complete assessment of susceptibility.

– **Incorporation of historical data**: Incorporation of historical data can significantly improve the accuracy of mass removal susceptibility models. However, most machine learning models are based on current or recent data, and the availability of historical data may be limited in some cases. Methods need to be developed to integrate historical data into landslide susceptibility models.

– **Development of interpretable models**: Although machine learning models can achieve high accuracy in identifying landslide susceptibility, some of these models can be difficult to interpret and explain. There is a need to develop interpretable models that allow decision makers to understand how the data is being used and how susceptibility identification is being performed.

The objectives of the work is to build a susceptibility model of the Chañaral province to identify the areas most exposed to landslide risk by using machine learning algorithms (SVM, RF, XGBoost and LR), and by comparing their performance; to


build an inventory of landslides in the study area through historical records and the analysis of satellite images and to determine the most relevant factors in susceptibility assessment by using indices based on information theory.

## 125  2  Materials and methods

### 2.1  Study area

The area of study is in the Province of Chañaral (26.4 °S), in the Copiapo region, Chile, between longitudes 70.7°E and 69.5°E and latitudes 26.2°S and 27°S (Fig. 1). In topographic and physical terms, the average altitudes range between 1000 and 1500 m above sea level. Geologically speaking, the area comprises 5 groups: (1) Quaternary period (Cenozoic era), (2) Paleocene
(Cenozoic era), (3) Cretaceous (Mesozoic era), (3) Middle Jurassic - Upper Jurassic (Mesozoic era), (4) Triassic - Lower Jurassic (Mesozoic era), (5) Devonian - Permian (Upper Paleozoic). Landslides have mainly occurred on these geological formations. The type of landslides found in this region correspond to alluviums and debris flows. According to the climatic classification, the area of study is in the transitional area between the hyper-arid and semi-arid zones (González, 2018). Rainfall averages 1.7 mm per year in the lower zone of the basin and can reach up to 52 mm per year in the upper zone, which
accumulates between June and August (Juliá et al., 2008) and the average temperature is between 10 and 20° C for the coastal and medium arid zones and -1.7° C in the high-altitude zone (Antonioletti et al., 1972).




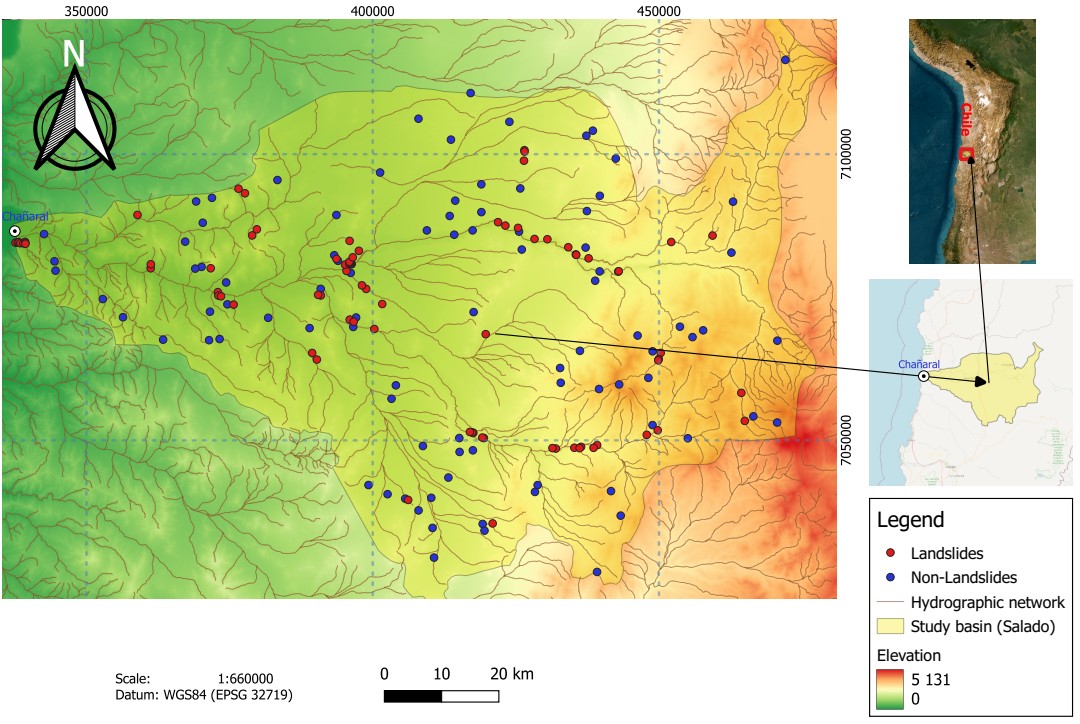

**Figure 1.** Map of the area of study. Extracted from (Tadono et al., 2016), © Google Maps 2023 and ESRI, 2023.

## 2.2 Landslide inventory map

To examine the correlation between the spatial prediction of landslides and the relevant influencing factors, it is imperative to consider the most recent landslide records available. Consequently, to establish a comprehensive and reliable inventory for the study area, information from previous studies will be used and subsequently verified by laboratory analysis. Location of landslides was recorded through the analysis of the Sernageomin database, bibliographic search and interpretation of aerial photographs, satellite images extracted from Sentinel Hub and the Copernicus repository, as well as the use of the Google Earth program. The information is managed together through the free software Qgis version 3.22 and scripts created in R that contain the necessary instructions for the geospatial processing of the data and the extraction of characteristics.

For the execution of the work, 86 locations of landslide events in the area of study were used, and another 86 locations were used as points where no landslides occur, to balance the data collection. In any way, having a small amount of data, it is necessary to resort to a cross-validation process, and thus obtain an average AUC that can be compared between the models used.



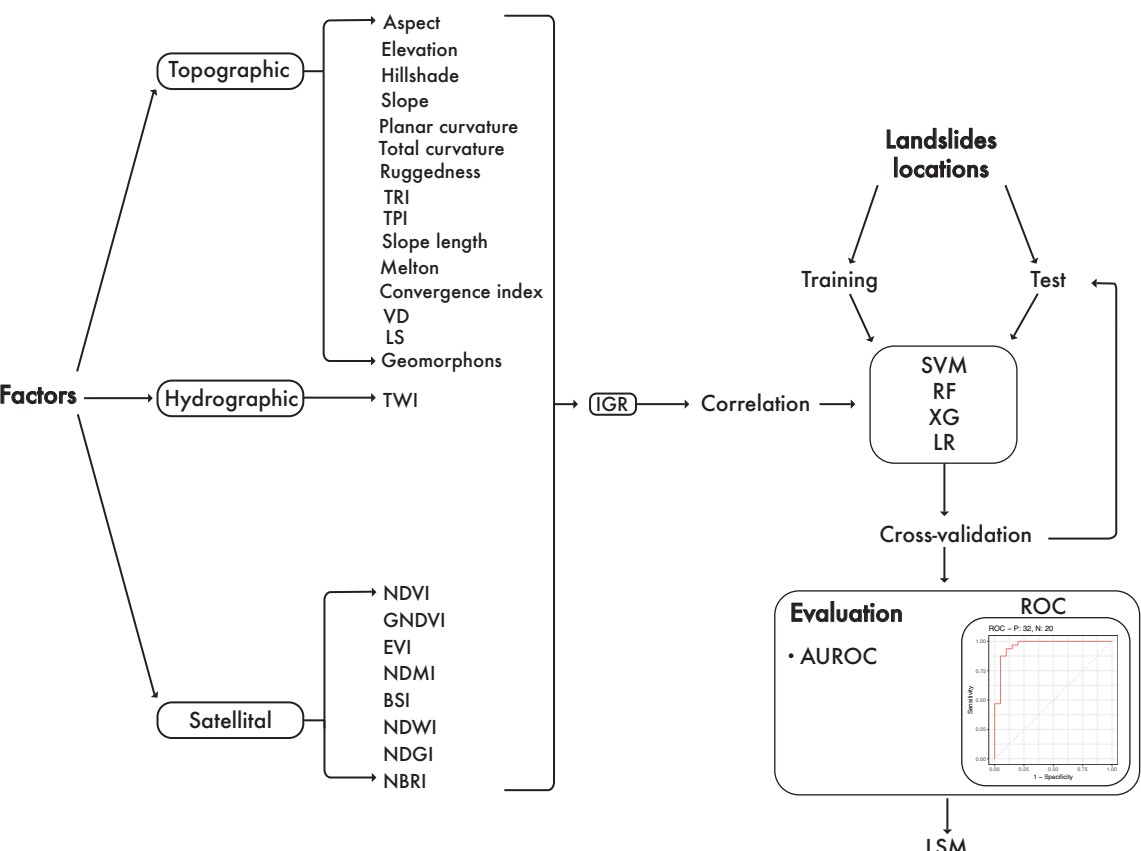

**Figure 2.** Experimental model.

## 2.3 Landslide conditioning factors

In this study, 23 factors associated with the landslides were used, which were obtained mainly through the DEM of the AW3D30 project (Tadono et al., 2016) and their respective analysis through R and multiple geoprocessing packages, in addition to satellite images received from the LANDSAT 9 campaign (Masek et al., 2020), which include eleven spectral bands that can be combined to identify characteristics in the ground. On the basis of the factors previously mentioned, the selected factors are the slope angle and orientation, the curvature, the elevation above sea level, the profile and plane curvature, the valley depth (VD),

the stream power index (SPI), the topographic wetness index, and the slope length. Additionally, the normalized indices NDVI, GNDVI, EVI, NDMI, BSI, NDWI, NDGI were obtained from images from February 2022. Then, for the machine learning analysis and the respective algorithms, the data relating to each category associated with each factor are included and then analysed using R.. Table I summarizes the factors and the types of classes into which they can be classified, as a reference. Also, figure 3 and 4 show thematic maps of all the factors presented in this study along with landslide inventory.




**Figure 3.** Thematic maps - 1.


**Figure 4.** Thematic maps - 2.





**Table 1.** LCF used in the study.

| Factors | Type |
|---|---|
| Aspect | Numerical |
| Elevation (m) | Numerical |
| Hillshade | Numerical |
| Slope (º) | Numerical |
| Total curvature | Numerical |
| Plane curvature | Numerical |
| TWI | Numerical |
| TRI | Numerical |
| TPI | Numerical |
| Slope length | Numerical |
| Melton | Numerical |
| Convergence index | Numerical |
| Valley depth | Numerical |
| LS | Numerical |
| Geomorphons | Categorical |
| NDVI | Numerical |
| GNDVI | Numerical |
| EVI | Numerical |
| NDMI | Numerical |
| BSI | Numerical |
| NDWI | Numerical |
| NDGI | Numerical |

**2.4   Factors selection using the Information gain ratio technique (IGR)**

Susceptibility assessment depends on the contributing elements. There are multiple techniques to determine the capacity of predictive elements that play a role in the occurrence of landslides, such as the gain ratio(Nithya and Duraiswamy, 2014), the significance of relief (Ahmad and Dey, 2005), and the information gain ratio (IGR) (Chapi et al., 2017). In this research, the latter has been chosen as the metric to quantify the predictive power of the contributing elements. The IGR methodology is

165 used to identify the most relevant elements among the 23 contributors previously discussed in the field of research.

Let consider $F$ as the data set used for training, containing an initial sample $n$. Let the set $n(M_i, F)$ represent the number of samples in the training set $F$ that belong to the class $M_i$ (which can be landslide or non-landslide). Consequently, the following equation can be established (Abedini et al., 2019b):





$$Info(F) = \sum_{i=1}^{2} \frac{n(M_i,F)}{|F|} log_2 \frac{n(M_i,F)}{|F|}(F) \tag{1}$$

Given the factors affecting the occurrence of landslides, the amount of information needed to divide $F$ in the series $(F_1, F_2, ..., F_m)$ is calculated through:

$$Info(F,E) = \sum_{j=1}^{m} \frac{F_j}{|F|} Info(F) \tag{2}$$

The calculation of the IGR index for a given factor is carried out in the following way:

$$IGR(F,S) = \frac{Info(F) - Info(F,S)}{Splitinfo(F,S)} \tag{3}$$

Where $Splitinfo$ means the information generated by partitioning $F$ of the training data into a subset of $l$, computed by:

$$Splitinfo(F,E) = \sum_{j=1}^{l} \frac{F_j}{|F|} log_2 \frac{F_j}{|F|} \tag{4}$$

## 2.5 Correlation calculation

In addition to the IGR, the Pearson correlation factor will be used to eliminate features that are correlated with each other, since this can cause noise and not contribute to the model's performance. This test is important to evaluate the dependence between conditioning factors. Generally, Pearson's correlation establishes the ratio between the covariance of a pair of factors and the product of their standard deviations (Dou et al., 2019).

## 2.6 Modeling using machine learning

Machine learning corresponds to an empirical approach for both classification and regression in non-linear systems. Such systems can be multivariable, involving literally thousands of variables. In machine learning, if there is sufficient data, a training data set is built covering as much of the system's parameter space as possible. Typically, a random subset of the data is set aside for completely independent validation. Machine learning is ideal for handling those problems where theoretical knowledge is still incomplete, but for which a certain number of meaningful observations and other data are available (Lary et al., 2016).

### 2.6.1 Support vector machine (SVM)

It corresponds to an algorithm based on statistical learning theory, used in regression and classification problems (Vapnik, 1999). In the work in progress, the classification mode will be used. The main characteristic of this method is that during the





learning process the algorithm transforms the initial space to a higher dimensional one, which allows the establishment of hyperplanes that are able to separate easily and thus classify new examples (Kavzoglu et al., 2015). In addition, it can work with nonlinear problems thanks to the incorporation of a Kernel, whose performance is controlled by the value $\gamma$ (Tien Bui et al., 2016). The model´s precision is also controlled by the C regularization parameter. Both parameters can be fine-tuned using the grid search technique (Kavzoglu et al., 2015) or using the random search.

### 2.6.2 Logistic regression

This method has been mainly used in the last decade in susceptibility assessment (Budimir et al., 2015), since it has proven to be very useful as base model when a new one is being tested (Chang et al., 2019). Logistic regression is the equivalent of linear regression, which uses a non-linear transformation to estimate class. It can calculate the weights of each conditioning factor as independent variables based on the binary dependent variable at a certain level of statistical confidence (Shirzadi et al., 2012). Advantages of the method include: (1) It does not require the data set to have a normal distribution, (2) The independent and dependent variables can be either continuous or discrete, and (3) It does not assume that the variables have the same statistics in their variances (Dou et al., 2019).

### 2.6.3 Random forest

The Random Forest (RF) is one of the most used methods in machine learning (Breiman, 2001). The model generates multiple classification trees to then obtain a final weighted score (Breiman et al., 2017). The algorithm adds diversity among classification trees by alternating data and further modifying the set of explanatory factors arbitrarily over the various processes of tree induction (Arabameri et al., 2020). The hyperparameters that are necessary for the growth of the tree are the number of trees $k$ and the number of predictive factors used to split the nodes ($m$). The OOB error (*out of bag*) is characterized as the percentage of the total number of objects that are misclassified, therefore it is a rational estimate of generalization error. The OOB error is estimated at the moment of building the model. In (Breiman, 2001) it is mentioned that the random forest creates a limiting value for the generalization error. Such error often declines as the number of trees grows. In turn, $k$ must be large enough to allow such convergence. The method calculates the value of the predictive variable by examining how much the error declines as the data are permuted for that variable while holding constant for the others. The growth in error corresponds to the value of the explanatory variable (Breiman, 2001). One of the main advantages of the random forest is its resistance to overtraining and the development of many trees where there is no risk of overfitting. Therefore, there is no need to rescale, transform or change the algorithm. For the predictors, the random forest is not too affected by outliers and deals missing values automatically (Crippen, 1990).

### 2.6.4 XGBoost

This method originated from the boosting tree gradient algorithm(Friedman, 2001). It uses regularized boosting techniques to reduce overfitting, thus improving the accuracy of the model. XGBoost it can scale in diverse scenarios, handle sparse





data, use scarce computational resources with high performance, have extensive and detailed documentation, and be simple to implement (Chen and Guestrin, 2016). This algorithm has won multiple contests (Chen and Guestrin (2016), Nielsen (2016)),

it has extensive hyperparameters that when synchronized substantially improve the model.

XGBoost is an extension of the gradient boosting algorithm. The main idea of a boosting algorithm is to combine several weak learners sequentially to achieve better performance (Hastie et al., 2009). The method uses several classification and regression trees (CART) and integrates them using the boosting gradient method. XGBoost is made up of three aspects that differentiate it from the other algorithms, being: (i) an objective function regularized for better generalization, (ii) a boosting

gradient tree for additive training, and (iii) a columnar subsampling to prevent overfitting (Chen and Guestrin, 2016).

## 2.7 Hyperparameters optimization

Hyperparameters correspond to the values that are setup before data training and generally affect the performance of predictions generated. These actions improve the performance by fine-tuning these hyperparameters (Todorov and Billah, 2022).

For the search of optimal hyperparameters, it is common in literature to use the grid search, which was used in this work.

**Table 2.** Hyperparameters used for SVM.

| Name | Lowest value | Highest value |
|---|---|---|
| Cost | 0.1 | $10^7$ |
| Sigma | 0.001 | 1 |

**Table 3.** Hyperparameters used for RF.

| Name | Lowest value | Highest value |
|---|---|---|
| num-trees | 1 | 1000 |
| mtry | 1 | 7 |
| alfa | 0 | 1 |
| max-depth | 0 | 100 |
| min-node-size | 1 | 100 |
| minprop | 0 | 0.5 |
| random-splits | 1 | 1000 |
| num-threads | 1 | 1000 |
| sample-fraction | 0.1 | 1 |
| seed | $-1000$ | 1000 |





**Table 4.** Hyperparameters used for XGBoost.

| Name | Lowest value | Highest value |
|---|---|---|
| col-sample | 0.3 | 0.7 |
| eta | 0.2 | 0.8 |
| gamma | 0 | 10 |
| max-depth | 1 | 1000 |
| min-child-weight | 0 | 8 |
| nrounds | 1 | 100 |
| subsample | 0.5 | 1 |

**Table 5.** Grid resolution.

| Model | Hyperparameters | Resolution | Configurations |
|---|---|---|---|
| SVM | 2 | 100 | $2^{100}$ |
| RF | 10 | 5 | $10^{5}$ |
| XGBoost | 7 | 5 | $7^{5}$ |

## 2.8 Cross-validation and model validation

Given the limited data, it is necessary to carry out a cross-validation process for the model validation. It refers to the process of repeatedly dividing the data set into a training set and a test set, where the former is used to fit a model, which is applied to the test set. When comparing the predicted values with the known values of the test set, it is possible to obtain a statement with reduced bias on the model's ability to generalize the model to unknown data. In this case, a 100-fold repeated 5-fold cross-validation is used, which means randomly dividing the data into five partitions to be used once as a test set. This ensures that each observation is used in the test set, which requires the fitting of five models. Subsequently, the process is repeated 100 times. In each iteration the cut of data shall be different. In summary, this leads to 500 models, where the average measure performance (in this case the AUC value) measures the overall model´s predictive power (Lovelace et al., 2019). When applying the 5-fold cross-validation method, it is equivalent to dividing the data set by considering 80% for training and 20% for validation/testing. Unlike traditional methods, each of the folds is used at some point for training and also for validation. Therefore, the ROC curves presented correspond to the averages obtained, considering that this process was repeated 100 times to achieve greater statistical robustness.

### 2.8.1 Model validation

Validation performance is a critical step within a modeling procedure; thus, several statistical indices have been suggested and used. In this work, the ROC curve will be used which is a basic measure in this type of evaluations (Pham et al., 2018). The plot is constructed with specificity and sensitivity on the x and y axis respectively (Pham et al., 2018), (Shirzadi et al., 2012).





Currently, the predictability of landslides in the respective area is examined by using a curve under the ROC curve (AUC) (Abedini et al., 2019b).

The statistic to be used for comparing the models corresponds to the average of AUC values obtained in the 500 iterations carried out through cross-validation.

Besides AUC (Area Under the Curve), the efficiency of the landslide models will be evaluated through statistical analysis by comparing the classification errors between the models. In general, a parametric test should be used for these cases. However, the values obtained from the error classification have a distribution that does not meet the normality assumption necessary for this type of test, which was tested by the test of Kolgomorov-Smirnov Berger and Zhou (2014), with a result lower than 260 0.05 for all models, which implies that the null hypothesis that values have a normal distribution is rejected. Furthermore, when performing the Box-Cox transformation it is also not feasible to normalize the models. Therefore, the Friedman non-parametric test will be used to determine statistical differences between the models. This test corresponds to the non-parametric equivalent of the ANOVA test and is used when the samples come from the same distribution and are paired and is used to determine whether the average of the populations is equal (Ostertagova et al., 2014). This test only shows the significant differences 265 between the models without judging pairs between two or more models. To discriminate between the models, the Nemenyi test will be used, which corresponds to a post-hoc test whose objective is to find groups of data that are different after a global statistical test (in this case the Friedman test) has rejected the null hypothesis that the performance of the models is the same. This test carries out pairs test to measure performance (Nemenyi, 1963).

Fig. 2 summarizes the design methodology respect to the procedures used this work.

## 270 3 Results

### 3.1 Selection of the landslides conditional factors

To determine which factors have contributed to the quality of the model, these were evaluated by the IGR technique in the area of study. Figure 5 illustrates the results of the IGR index for the 12 factors selected in the study area, all of which are greater than 0. The findings show that the Valley Index (VD) has the most significant predictive capacity for the model. On 275 the other hand, the Melton index have the lowest value. Other factors, which include TWI, NDGI, TPI significantly contribute to the landslide model. In contrast, the 12 factors selected (aspect, elevation, hillshade, total and planar curvature, Factors such as Slope Size, TRI, Ruggedness, NDMI, NBRI, BSI and LS Factor have a merit value equal to 0, which results in their exclusion from the modelling process. This is due to the detrimental effect of introducing noise into the model, which reduces the predictive ability of the model (Tien Bui et al., 2016).

Additionally, correlation among the 12 factors chosen in the previous stage is measured removing from the analysis those that have a lower impact and related with other factors that have greater impact on the model. Under this perspective, the Melton factor, Geomorphons, NDVI, NDWI and GNDVI are excluded from the analysis, so that seven factors are finally used to build the model: VD, TWI, NDGI, TPI, convergence index, planar curvature and EVI.





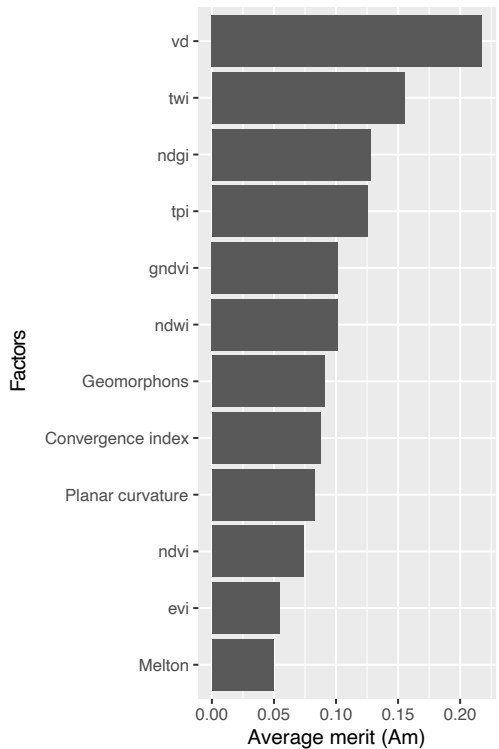

**Figure 5.** Predictive power of factors according to IGR.





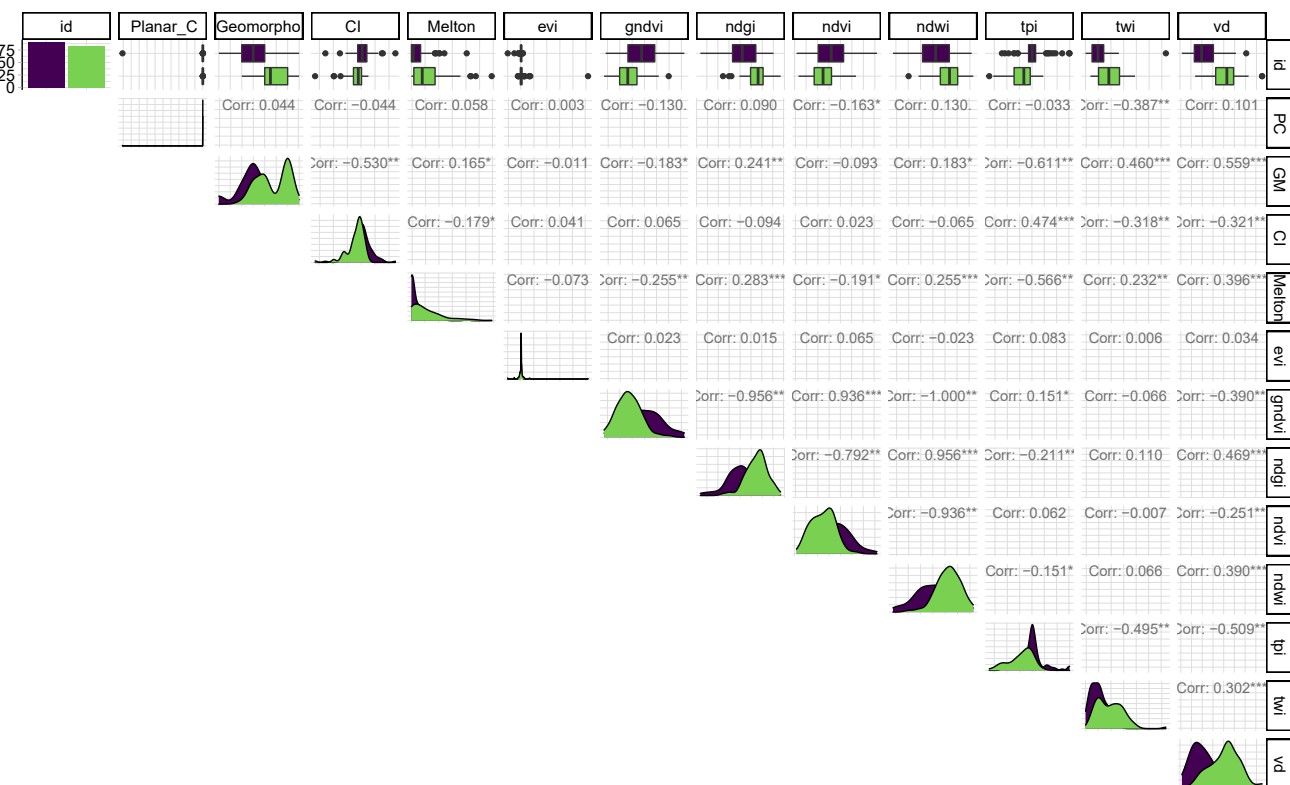

**Figure 6.** Summary chart – Correlation.

**Table 6.** Parameters of landslide modelling algorithms in this study.

| Algorithm | Parameter |
|---|---|
| SVM | C = 0.9296, sigma = 0.0021, kernel = RBF |
| RF | num-trees = 250, mtry = 2, alpha = 1, max-depth = 0, min-node-size = 1, minprop = 0.125, num-random-splits = 1, num-threads = 500, sample-fraction = 0.7750, seed = 1000 |
| XGBoost | colsample-bytree = 0.5, eta = 0.65, gamma = 5, max-depth = 250, min-child-weight = 2, nrounds = 100, subsample = 0.875 |
| LR | No hyperparameter optimization |

## 3.2 Models analysis

In this study, machine learning models were implemented by using the R programming language through the mlr3 package
Lang et al. (2019), which is a complete machine learning models analysis ecosystem. Optimal values of the hyperparameters
obtained through the method are shown in table 6.





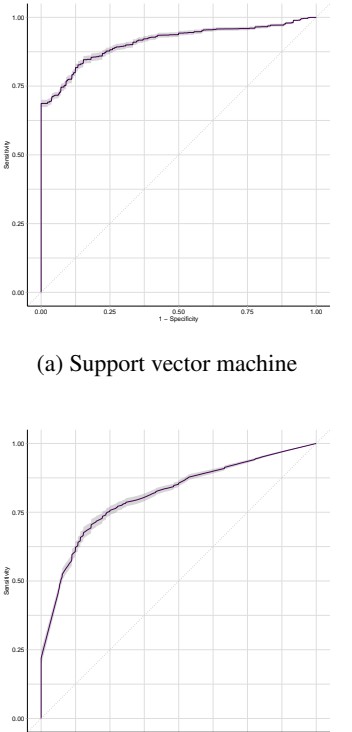

(a) Support vector machine

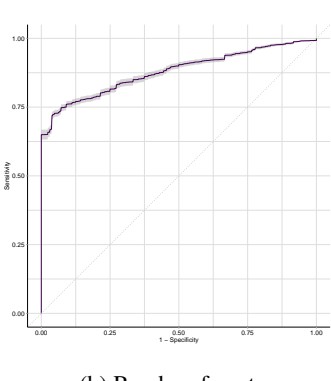

(b) Random forest

(c) XGBoost

(d) Logistic regression

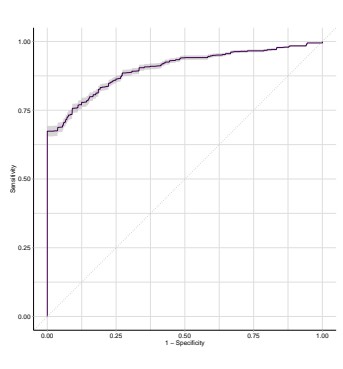

**Figure 7.** ROC curves after cross-validation. Test set

## 3.3 Model performance and validation

In the model´s evaluation, factors including the average ROC curve among all iterations produced by the cross validation,
and the respective area created under the curve AUROC, are used. AUROC values vary between 0.5 and 1, where 0.5 implies
having a precision identical to a model set randomly, while 1 represents the optimal model with the maximum area under the
curve. Figure 7 summarizes the ROC curves for the test set, and figure 8 shows the difference between the AUC of the training
set and the test set for all models. This shows that in the training set the solution was overfitted.

Taking into consideration the hyperparameters shown in table II and using the factors that contain the most information,
SVM models, logistic regression, RF and XGBoost are obtained. The AUC value average result is shown in table III using
cross-validation. Figure 9 also shows a box plot that allows comparing the model´s statistical distribution with respect to the
classification error. This graphic shows the values obtained for the process repeated 500 times (5-fold cross-validation with
100 repetitions), including the mean and the value distribution. This metric is obtained in the same process as the AUC.


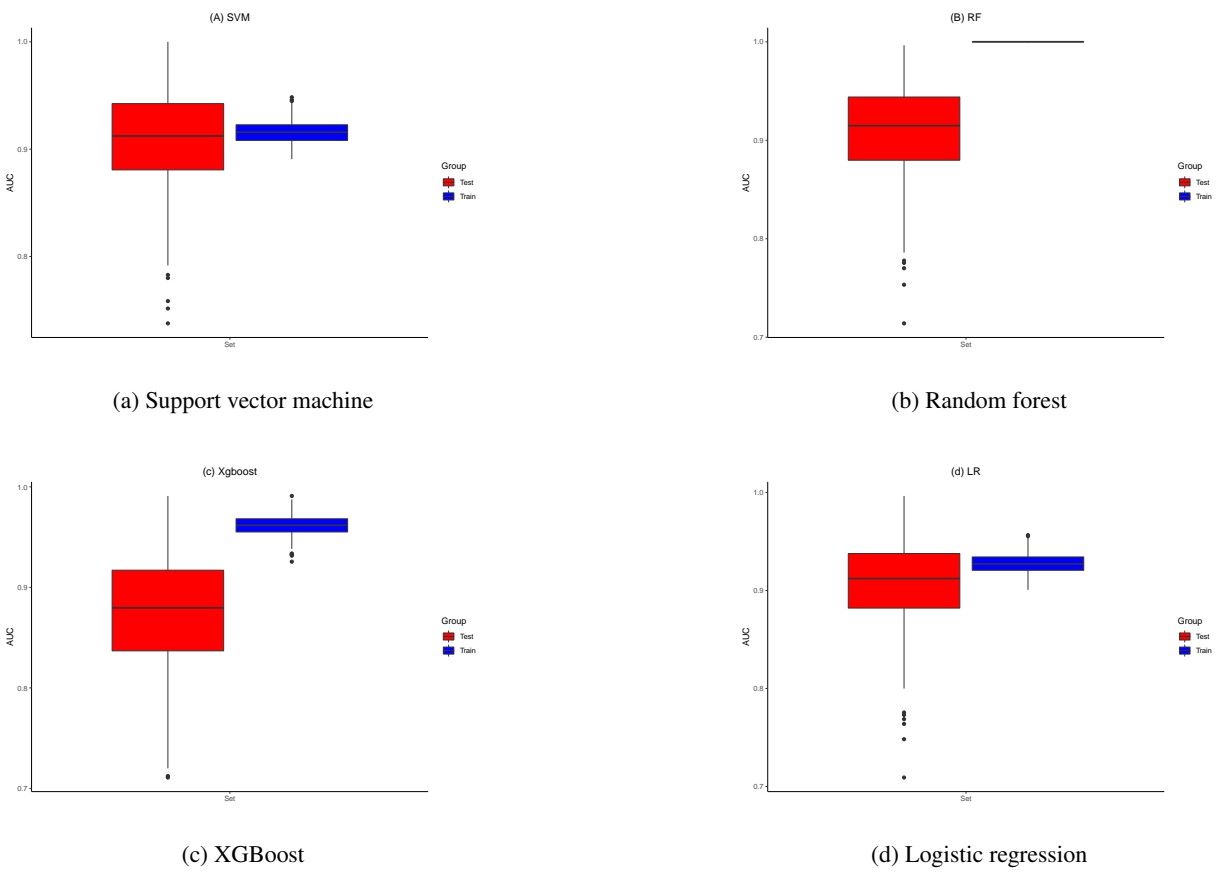

(a) Support vector machine

(b) Random forest

(c) XGBoost

(d) Logistic regression

**Figure 8.** Boxplot of AUC for training and test set. In red the test set and in blue the training set

For the statistical analysis, table 8 summarizes the results of Friedman´s overall test considering the 500 observations of
the classification error for each model. Finally, table 9 shows Nemenyi´s post-hoc test results allowing the comparison among
each of the models.

**Table 7.** Average AUC for studied models

| SVM | RF | XGBoost | LR |
|------|------|---------|------|
| 0.9089 | 0.9095 | 0.8757 | 0.9085 |




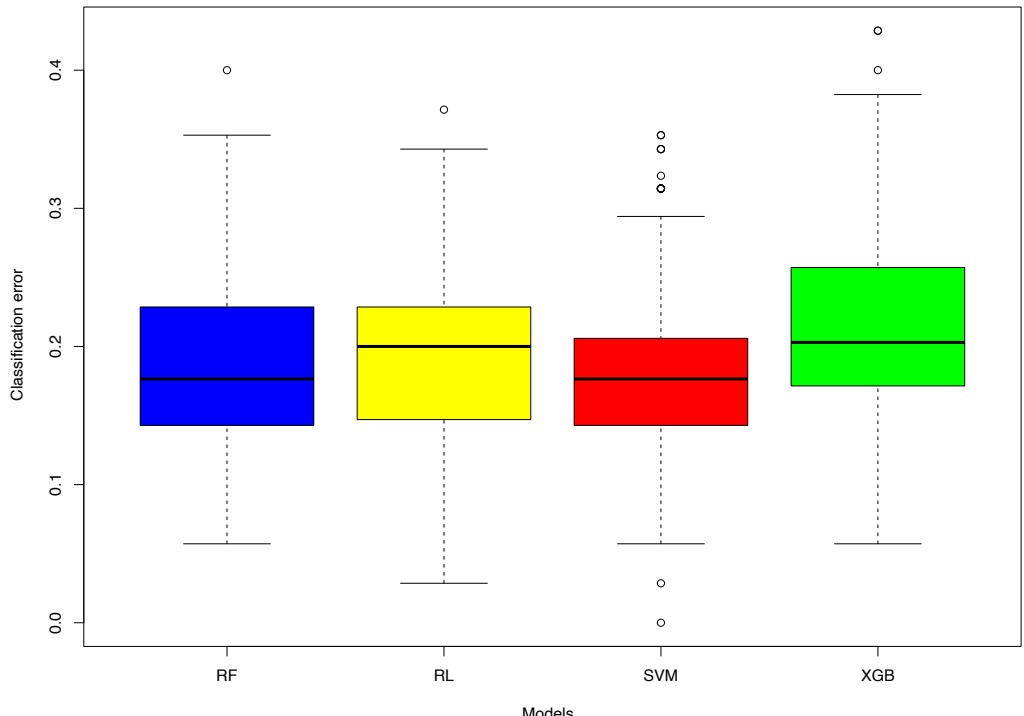

**Figure 9.** Comparison of classification error values for the different models

**Table 8.** Friedman's test results for the four models

| Number | Statistic | p-value |
|--------|-----------|---------|
| 1 | 100 | 1.29e-21 |

**Table 9.** Comparative analysis in pairs for the model of the four susceptibilities by means of the Nemenyi test.

| Number | Compared pair | p-value | Significant |
|--------|---------------|---------|-------------|
| 1 | SVM vs RF | 0.245 | No |
| 2 | SVM vs XGBoost | 6.9e-14 | Yes |
| 3 | SVM vs RL | 0.018 | Yes |
| 4 | RF vs XGBoost | 7.8e-09 | Yes |
| 5 | RF vs RL | 0.711 | No |
| 6 | XGBoost vs RL | 3.4e-06 | Yes |





## 3.4 Susceptibility Maps

After evaluating the performance of the four prediction methods, the respective landslide susceptibility maps were made. To do so, the following steps are taken:

– Ten million points are generated in the basin polygon, evenly distributed.

– At each of these points, the values of the factors causing the landslides (VD, TWI, NDGI, TPI, EVI, Convergence Index and planar curvature) are calculated.

– Using the machine learning models, the landslide susceptibility indexes are calculated for each point.

– The points are transformed into a georeferenced raster file.

– The values obtained in step three are reclassified into regular intervals ranging from 0 to 1, using the following labels: very low susceptibility, low susceptibility, middle susceptibility, high susceptibility, and very high susceptibility.

Figures 10-13 show the maps generated by the four models under study. As seen in the figure, the four maps indicate similar areas of susceptibility. The main difference is that both the random forest and the XGBoost show greater detail than the SVM and the logistic regression. For the calculation of the thresholds, it was decided to have equal intervals and then divide them

into the respective labels, in this way:

– Very low susceptibility: 0 – 0.1987

– Low susceptibility: 0.1987 – 0.3943

– Middle susceptibility: 0.3943 - 0.5898

– High susceptibility: 0.5898 – 0.7854

– Very high susceptibility: 0.7854-1


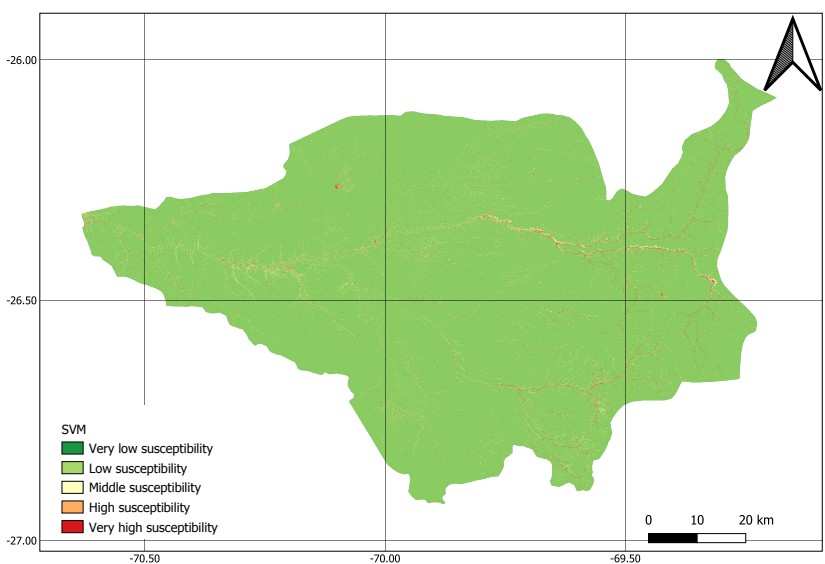

**Figure 10.** Susceptibility map – SVM

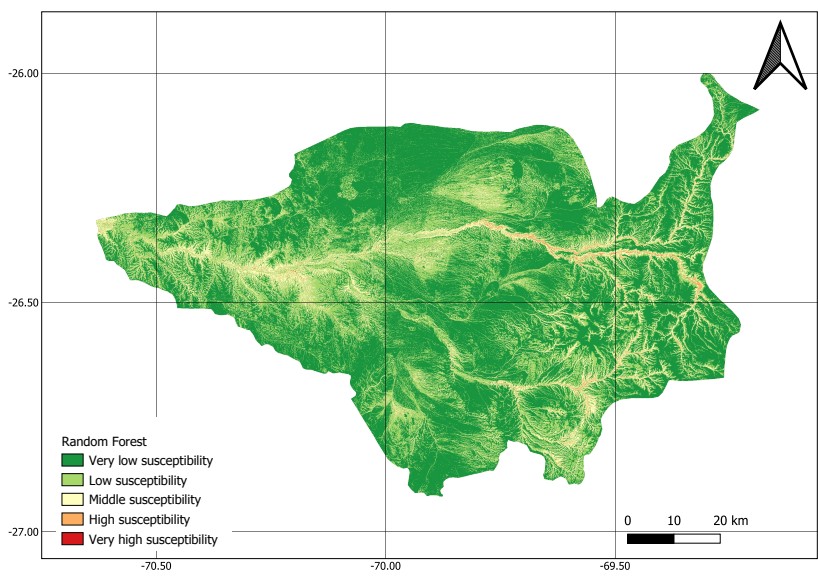

**Figure 11.** Susceptibility map – RF



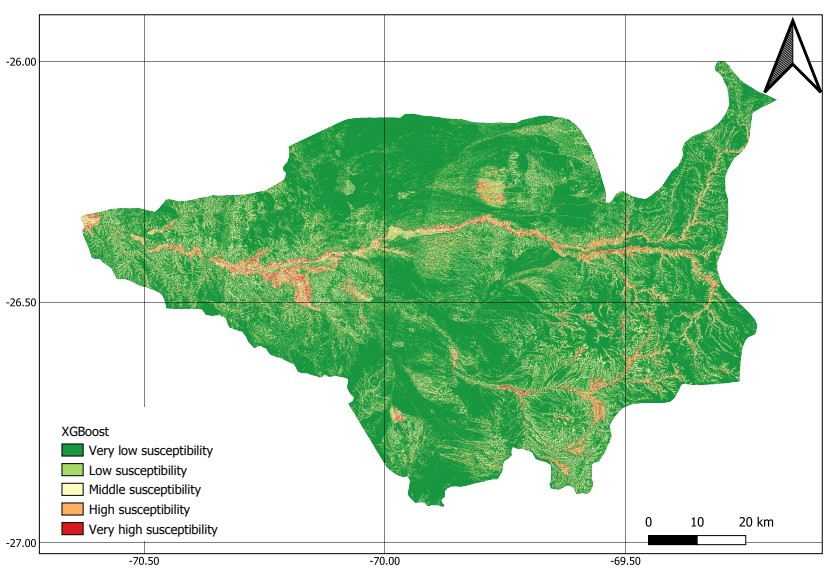

**Figure 12.** Susceptibility map - Xgboost

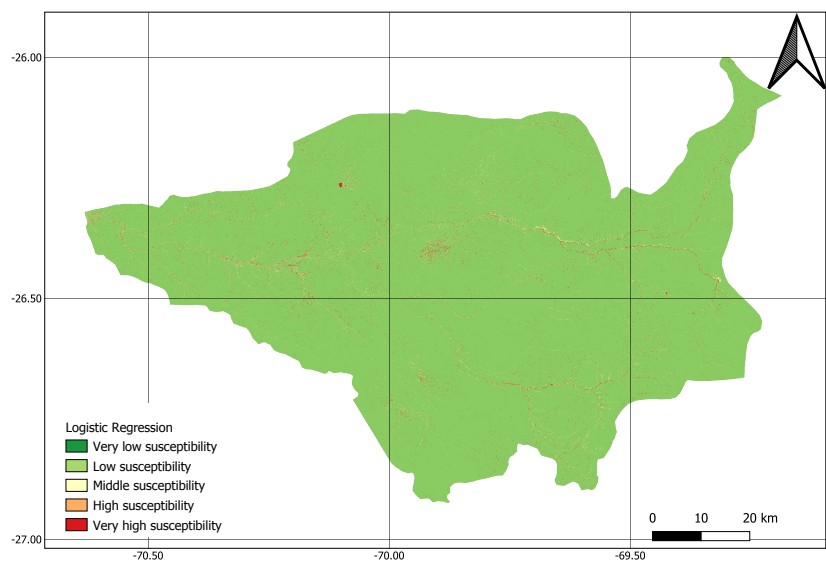

**Figure 13.** Susceptibility map - Logistic regression



## 4   Discussion

The spatial prediction of landslides is considered to be one of the most complex tasks in natural hazard risk assessment. Despite the fact that numerous methodologies have been proposed, the accuracy of the predictions is still a controversial issue citepakgun. The development in the field of machine learning and the GIS platforms has led to the development of many new

techniques, and methods. However, the further exploration of new methods is still necessary.

This research addresses this issue by evaluating and comparing four machine learning techniques. In general, SVM and RF outperform the other models in terms of classification effectiveness. In terms of hyperparameter calibration, the available computational resources have been used to perform a grid search. In the case of RF and Xgboost, these algorithms need to adjust a larger set of parameters..

In regard to AUC (Table 7), RF obtains the highest value. However, relying only on this metric might not be the optimal strategy, since higher values of AUC do not necessarily guarantee a higher spatial accuracy of the models (Aguirre-Gutiérrez et al., 2013). Therefore, other additional metrics of statistical evaluation are needed. The Friedman statistical test showed that there are significant differences among the methods. Then, using the Nemenyi pairwise test, it can be seen, that although RF, SVM and RL have significant statistical differences with XGBoost, the relation among them is not so clear, since RF has no

differences with SVM or RL, while these two do.

The machine learning models are suitable for solving the studied problem, since they are able to handle the complex relationships between LCFs and removal susceptibility and to be robust in noisy environments (He et al. (2012), Wu et al. (2013), Huang et al. (2020)). The algorithms presented in this paper have been widely used in the literature for the generation of removal susceptibility maps. SVM has obtained AUROC values ranging from 0.768 to 0.946 (Abedini et al. (2019b), Zhao et al.

(2022), Huang et al. (2020), Huang et al. (2022), Chen and Guestrin (2016)). Logistic regression, which is mainly used as a benchmark with which it is possible to make a comparison with other models, has obtained AUROC values ranging from 0.792 to 0.934 (Zhao et al. (2022), Tsangaratos and Ilia (2016), Bruzón et al. (2021), Zhu et al. (2020), Huang et al. (2022)). On the other hand, XGBoost, although it has been used in fewer publications than the other algorithms, has obtained promising results: in Can et al. (2021) it obtained an AUROC of 0.96, while in Bruzón et al. (2021) it obtained an AUROC of 0.979. Finally,

RF, which almost always obtains outstanding results in this problem, has an AUROC ranging from 0.9 to 0.985 (Zhao et al. (2022), Tsangaratos and Ilia (2016), Bruzón et al. (2021), Arabameri et al. (2020), Huang et al. (2022)), which is consistent with the results obtained, which are also supported by findings from previous studies (Pourghasemi and Rahmati (2018), Ali et al. (2021), Zhao et al. (2022)). One of the advantages that RF has in conjunction with XGBoost is that both are immune to multicollinearity that can occur due to the presence of multiple topographic derivatives as conditioning factors (Kotsiantis

(2013), Piramuthu (2008), Can et al. (2021)). Other advantages of RF are that it does not require assumptions on the statistical distribution in the conditioning factors, it takes into account interactions and nonlinear characteristics among the variables, and the ability to provide information on the influence of each variable in the final model (Catani et al. (2013), Pourghasemi and Rahmati (2018), Tsangaratos and Ilia (2016)). The differences between the three models lie mainly in the fact that the principles they use to generate predictions are different. The SVM is able to map low-dimensional features to high-dimensional



spaces using a kernel to find a characteristic hyperplane to maximize the categorical space. The problem with this method
       is that the corresponding mapping may be poor for the prediction in question. The RL characterizes the spatial relationship
       between the removal events and the conditioning factors looking for the best fitting algorithm. However, it is very sensitive to
       multicollinearity, which limits its performance (Huang et al., 2022). On the other hand, XGBoost does not achieve the perfor-
       mance described in the literature, being outperformed by the rest of the models. This could be explained by the fact that both

RF and SVM need few hyperparameters to be tuned, while XGBoost needs more to achieve its best performance. Moreover, it
       is more difficult to work with if there is a lot of noise in the data. Also, XGBoost can lead to overfitting if the number of trees
       is not carefully controlled (Abedi et al., 2022).

       The choice of conditioning factors is a key aspect that influences the quality of susceptibility models (Costanzo et al., 2014).
       Although various methodologies for selecting factors have been proposed, including linear correlation (Irigaray et al., 2007)

and the Kolmogorov-Smirnov test (Costanzo et al., 2014), there is still no universal criterion for making these selections, and
       the issue remains a topic for debate (Tien Bui et al., 2017). In general, topographic, geologic, soil, hydrologic, geomorphologic,
       and anthropogenic factors have been accepted in the literature for most susceptibility models. In some cases, factors that do not
       have predictive capability cause noise, affecting the quality of the model. In addition, it is important to eliminate those factors
       that have a high correlation index between them, to be able to apply cross-validation.

Using the information gain ratio (IGR) it is possible to measure the amount of information that a factor can provide to
       a model as a conditioning factor. In this research, out of the 23 factors initially considered, 12 of them show no predictive
       contribution to the models. This is attributed to the fact that the dispersion of locations within the categories of these factors is
       relatively even compared to the others. As a result, there is no significant reduction in the entropy of the two categories when
       they are associated with these factors. In any case, in order to reduce bias, it is suggested to base the result on a voting system

among different models (Xu et al., 2014).

       The results obtained (AUC greater than 0.9) confirm what has been shown in the literature, in the sense that both RF and
       SVM are algorithms that perform well when working with landslide susceptibility. However, one notable disadvantage of the
       RF model is its inability to calculate the relative importance of each subclass of the LCF, which limits its effectiveness in
       certain situations. In contrast, SVM uses "support vectors" and is better suited to sparse datasets, with the added advantage of

having non-linear kernel functions, but has a tendency of overfitting (Sharma et al., 2021). For this reason, we work with the
       cross-validation methodology to prevent the characteristic overfitting.

       In this work, two factors that have not been considered in the literature are included, the NDGI (Normalized Difference
       Glacier Index) and the EVI (Enhanced Vegetation Index), instead of the NDVI, which has been widely used. However, it has
       important limitations, such as its dependence on the daily time in which the aerial images are taken, since it does not correct for

changes in the angle of solar incidence. Therefore, this index produces inaccurate results. In this sense, EVI, which is calculated
       similarly to the NDVI, uses additional wavelengths to correct the NDVI inaccuracies. This corrects for variations in the solar
       angle, atmospheric distortions caused by airborne particles and land cover signals under vegetation. On the other hand, the
       NDGI, which has mainly used for glacier characterization, has a high predictive value for the susceptibility estimation, given
       by the IGR, so it also replaces the NDVI. NDGI uses spectral bands corresponding to green and red, so this would imply that


landslide and non-landslide areas create contrast between these wavelengths. Therefore, it is suggested to use these indices in areas similar to the studied in this work.

It is also novel that the "Valley Depth" (VD) index is the one that provides the most information for the model. This characteristic corresponds to the vertical distance to a base level of the hydrographic network. The algorithm that calculates this index consists of two steps, which involve the interpolation of the elevation of the base level of the hydrographic network,

and the subsequent subtraction of this base level from the original elevations (Conrad et al., 2015). This implies that the landslide and non-landslide sites in the area share similar values of VD respectively. This aspect is important for morphologies such as that of the Salado River basin, which has a marked slope at the geographic transition as it crosses from the foothills to the intermediate depression and has a "funnel" shape (González, 2018).

Among the factors studied in this work, two stand out with respect to the others in terms of their influence on the model:

the valley depth index (VD) and the TWI. A high valley depth index may be related to a high susceptibility to landslides due to the steep topography and abrupt relief present in the study area, which may favor the occurrence of gravitational processes and increase the erosion rate on the slopes, while a high TWI indicates a saturated soil, which implies an increase in the susceptibility to landslides.

This study is novel in two respects from a machine learning perspective in that:

– A 5-fold cross-validation with 100 replicates is used to calculate the prediction metrics, while most studies use a static data partition to then calculate the indexes of interest. This methodology does not deal with the stochastic nature of the problem, so applying cross-validation with repetitions allows obtaining more robust results.

– According to the current state of the art, before performing the analyses, the natural break method (natural Jenks break) is used to calculate the susceptibility and the respective factors, which seeks to normalize the data by reducing the variance

and creating categories given by numerical intervals. However, there are several problematic issues when converting continuous data to categorical data. First, it is highly unlikely that the underlying trend is consistent with the new model. Second, when a true trend exists, discretizing the data will make it more difficult to do an effective job because the nuances of the data are removed. Third, there is no objective justification for the chosen cut-off points. Lastly, when there is no relationship between the outcome and the predictor, the probability of finding an erroneous trend increase (Kuhn

and Johnson, 2020). Therefore, continuous values are used in this work, which results in a more detailed production of susceptibility maps, especially for RF and Xgboost.

In summary, the novelty of this study consists of applying repeated cross-validation to obtain the metrics of the models, and the use of Valley depth index, NDGI and EVI to construct the susceptibility models. Also, other novelty is the use of the MLR3 package in solving the machine learning problem, and the combination with other geospatial packages in R in order to produce

the susceptibility maps.

The contributions to agriculture are varied, since although the area is not characterized as an agricultural zone as a result of intense mining activity, new productive poles are being generated in the region, such as jojoba production Menchaca (2007). Therefore, these susceptibility maps can help identify safe zones for agriculture, risk management and implementation of





preventive measures (e.g. soil conservation), and cost reduction. It should also be noted that this study is the first of its kind to
be carried out in this region.

Although there are not many susceptibility studies using machine learning in regions similar to the one studied, there is a
precedent in Peru: in Bueechi et al. (2019), they use a logistic regression model to calculate susceptibility in the Cordillera
Blanca, achieving an AUC of 0.75. The region in question presents topographic similarities with the Salado Basin, so the model
built in this study may have promising results in that area.

The applicability of the proposed model is determined by the climatic, topographic and the morphometric characteristics of
the study area. Under that perspective, the model can be expected to be suitable in areas worldwide that is a semi-arid zone,
with a variable topography and a Mediterranean climate with a prolonged dry season, in addition to having narrow and deep
valleys, where the maximum susceptibility is concentrated. Examples of these zones are the following:

– **Colca Valley, Peru**: This region is located in southern Peru and has a rugged topography with narrow and deep valleys.
  The climate is semi-arid with a prolonged dry season and has geomorphological characteristics similar to those of the
  Salado Basin.

– **Indo Valley, Pakistan**: This valley is located in northern Pakistan and is a mountainous region with deep, narrow valleys.
  The climate is arid with a prolonged dry season and the region has a geomorphology similar to the study zone.

– **Colorado River Valley, United States**: This region is located in the southern part of the state of Colorado and in northern
  New Mexico. It is a semi-arid area with a rugged and mountainous topography, and has narrow and deep valleys similar
  to those of the Salado Basin.

## 5  Conclusions

In activities such as urban planning, mining, land and soil management or agricultural production, the evaluation of landslide
susceptibility is a crucial task for decision makers and authorities, especially in areas that have been affected in the past by
this type of event, such as the Salado river basin, III Region, Chile. Therefore, there is a need to have an updated map with
the highest level of accuracy. That is why this study presents an extensive comparison and evaluation of four machine learning
models (Random forest, Support vector machine, Xgboost and Logistic regression). To train and validate this methodology, 86
locations identified as landslides and 86 locations identified as non-landslides were used, in addition to using 23 conditioning
factors, of which 7 were chosen using feature selection techniques like IGR (information gain ratio) and Pearson's correlation.
The models were tested using a 5-fold cross-validation repeated 100 times. The metrics used correspond to the area under the
AUC curve and the respective ROC curve to measure the level of accuracy of the models, and the classification error, used to
determine whether there are significant differences between models. Our results indicate that the RF and SVM models obtain
the highest AUC indices, with 0.9095 and 0.9089, respectively. Furthermore, the non-parametric statistical tests indicated
that there are no significant statistical differences between them. Finally, the maps for all models are generated. On the other
hand, the IGR indicates that the most relevant factor for the calculation of susceptibility in this area corresponds to the valley





depth index (Valley Depth), which indicates that this factor can be used in the measurement of susceptibility of landslides in geographic areas similar to the one studied, with narrow and deep valleys.

The main limitations found in this work are the following:

- Sample size limitation: The amount of data used in this study may limit the generalization of the results to other areas, and prevent reaching higher precision values.

- Integration of other factors: The study considers only topographic, hydrological and satellite factors for its analysis. Subsequent studies could include anthropogenic factors (urbanization or mining activity), geological or infrastructure factors in their analysis.

- Spatial resolution: The area studied in this work is considerably larger than those generally used in this type of study. It is suggested that future studies be limited to a sub-basin of interest and work with a resolution greater than 30 m/pixel.

These findings provide valuable perspectives for informed decision-making and policy formulation in landslide-prone regions. Overall, our study highlights the potential of machine learning models, particularly SVM and RF, for accurate and reliable landslide susceptibility mapping, which can aid in identifying high-risk areas and implementing effective mitigation strategies, which is useful for stakeholders and land-planning authorities.

*Code availability.* The code presented in this study are available on request from the corresponding author.

*Data availability.* The data presented in this study are available on request from the corresponding author.

*Author contributions.* The following statements should be used "Conceptualization, F.Parra; methodology, F.Parra. and M.Chacón; software, F.Parra.; validation, F.Parra., M.Chacón and M.Marín; formal analysis, F.Parra. and M.Chacón; investigation, F.Parra.; resources, F.Parra.; data curation, F.Parra. and J.González.; writing—original draft preparation, F.Parra.; writing—review and editing, F.Parra., M.Chacón and M.Marín; visualization, F.Parra.; supervision, M.Marín.; project administration, M.Marín.; funding acquisition, M.Marín. All authors have read and agreed to the published version of the manuscript.

*Competing interests.* The authors declare no conflict of interest.

*Acknowledgements.* This work has been partially funded by the Chilean Agency for Research and Development (ANID) under grant Basal Centre CeBiB code FB0001.





We would like to thank the graphic designer Camila Vargas, who was in charge of the realization of several figures in the manuscript.



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
