# Peer review of "Modeling and evaluation of the susceptibility to landslide events using machine learning algorithms in the province of Chañaral, Atacama region, Chile"

_Natural Hazards and Earth System Sciences, 2023_

## Community Comment (CC3)

*<<Our focus is on the application of machine learning algorithms for the generation of landslide susceptibility maps, which may differ from traditional studies>>*

No previous work attempted in the region can be considered traditional since they explore for the first time what are (i) the conditioning factors to be used in this arid region (year 2020, NHEES) and, (ii) are based on very novel remote sensing applications of SAR C-band satellites which are open access and I strongly recommend the authors to consider in future resubmissions (2020, ESPL; 2020; Natural Hazards). It's a close call since I am the first author, but you can also rely on other works using similar approaches (Castellanezzi et al., 2023 in Australia; Botey i Bassols et al., 2023 in the Salar de Atacama; Olen and Bookhagen, 2020 in Argentina among other references therein that you might fing useful too). There is also the work of Olivares et al. (2023) as I discussed before which might also be a good starting point to your goal of defining susceptibility maps in the Atacama.

*<<Our approach allows us to identify patterns and relationships>>*

The authors now claim to identify patterns and relationships. I have navigated the paper throughout and any physical processes, nor landslides triggering analysis is presented in the sense of giving clues about either the triggering or the possible feed-backs between the studied landscape features. If when authors claim 'patterns' they speak about areas under high susceptibility, again, previous work in the area proves them wrong. At least authors should discuss why their maps show differences with available literature in the area.

*<<The selection of conditioning factors was based on the Information Gain Ratio (IGR) technique, which quantifies the predictive power of the contributing elements. We selected the most relevant elements among the 22 contributors based on this methodology. >>*

It is not under discussion the utility of IGR or any other predictive technique. The problem, as I mention in my previous comment, is that this investigation starts from the wrong selection of conditioning factors by the authors. This selection was done by (I guess) a literature review conducted by the authors and there is where all the problems that this research has starts. You need to better understand, probably by doing a more thorough literature review (I just gave you some references in my previous comment), that studying arid regions is not a novel area and therefore it can be considered a traditional topic in geomorphology (see literature in the Sonoran Desert at the USA 1950's, etc). I believe a deeper review can be done here if the aim of the authors is to provide the basis of future research of susceptibility to landslides in arid regions since they mention '' *we believe that our findings provide a valuable basis for future research in this area.* '' many times in their response to my previous comment.

This paper now has a lot of conditioning factors selected under unclear criteria (not well justified or cited) and which have been proven wrong in recent and traditional literature in the region and in other arid regions of the world. This needs to be amended.

*<<The four models we use are widely recognised for their effectiveness in a variety of machine learning tasks, and cross-validation is standard practice for assessing model accuracy. >>*

The accuracy assessment it is not a problem for me since its based in wrong theoretical assumptions (bad choice of conditioning factors). So, I cannot see how this is relevant at this point.

<<It is important to note that although there are other studies that have reported landslides, flash floods and other runoff-related hazards in the same location, these studies may have used different methodologies and criteria to identify and classify these events. Therefore, it is not always possible or appropriate to directly combine these datasets with our own. >>

You give a dataset which relies on observations done near the main roads, villages, mining districts. This is typical for this region and can be a limitation. However, many authors have done efforts to overcome this by providing regional observations (see Wilcox et al., 2016; Tapia et al., 2018; Cabré et al. 2022). The combination with other datasets is possible because although using hydraulic and remote sensing methods, the works from Wilcox and Cabré use a lot of field data (see figure 2 in Cabré as an example).

Thank you for the attached figure of susceptibility map. I believe that authors may have come to a similar map by only doing a sort of slope thresholded map here. This makes me refer the authors to another work of myself (https://doi.org/10.1002/esp.4868) where we clearly showed (only 70kms south) that slope is not controlling rainfall-triggered hazards in this region. This might sound contra intuitive for unaware readers, but gentle surfaces are the ones that remarkably are more impacted during rainstorm events in this region. This can be explained because gentle slopes allow a great and better development of alluvial cover and thus make sediment available at any storm.

Other refs:

https://doi.org/10.3390/rs15123034

https://doi.org/10.1029/2019JF005141

https://doi.org/10.1016/j.rse.2023.113546

---

## Author Comment (AC5)

| Random Forest | |
|---|---|
| | Very low susceptibility |
| | Low susceptibility |
| | Middle susceptibility |
| | High susceptibility |
| | Very high susceptibility |